# A Comparative Study of Time Series Anomaly Detection Models for Industrial Control Systems

**DOI:** 10.3390/s23031310

**Published:** 2023-01-23

**Authors:** Bedeuro Kim, Mohsen Ali Alawami, Eunsoo Kim, Sanghak Oh, Jeongyong Park, Hyoungshick Kim

**Affiliations:** 1Department of Electrical and Computer Engineering, Sungkyunkwan University, 2066 Seobu-ro, Jangan-gu, Suwon-si 16419, Gyeonggi-do, Republic of Korea; 2Department of Computer Science and Engineering, Sungkyunkwan University, 2066 Seobu-ro, Jangan-gu, Suwon-si 16419, Gyeonggi-do, Republic of Korea

**Keywords:** anomaly detection, intrusion detection systems, industrial control systems, deep learning model, unsupervised learning

## Abstract

Anomaly detection has been known as an effective technique to detect faults or cyber-attacks in industrial control systems (ICS). Therefore, many anomaly detection models have been proposed for ICS. However, most models have been implemented and evaluated under specific circumstances, which leads to confusion about choosing the best model in a real-world situation. In other words, there still needs to be a comprehensive comparison of state-of-the-art anomaly detection models with common experimental configurations. To address this problem, we conduct a comparative study of five representative time series anomaly detection models: InterFusion, RANSynCoder, GDN, LSTM-ED, and USAD. We specifically compare the performance analysis of the models in detection accuracy, training, and testing times with two publicly available datasets: SWaT and HAI. The experimental results show that the best model results are inconsistent with the datasets. For SWaT, InterFusion achieves the highest F1-score of 90.7% while RANSynCoder achieves the highest F1-score of 82.9% for HAI. We also investigate the effects of the training set size on the performance of anomaly detection models. We found that about 40% of the entire training set would be sufficient to build a model producing a similar performance compared to using the entire training set.

## 1. Introduction

Industrial control systems (ICS) are automatic systems and associated instrumentation used to control and monitor critical infrastructures such as power plants, water treatment, smart factories, and many other facilities. Recently, ICS has been connected to the internet to manage ICS more conveniently and reduce the cost of the industrial process. However, connecting ICS to the internet increases harmful threats to critical infrastructures [1,2]. For instance, attackers can compromise ICS by obtaining network control through a social engineering attack on the system manager [3] or exploiting the security vulnerability of the legacy physical device connected with the network [4]. The attacks on ICS are increasing each year, which would impact human lives and economic losses [5]. For this reason, building a system that can detect cyber-attacks is essential.

Using an intrusion detection system (IDS) is an effective method to defend ICS against cyber-attacks. Recent research has explored the development of IDS mechanisms specifically targeted for ICS. An IDS monitors events and detects suspicious activities and intrusions. Existing approaches have been categorized into three types: (1) knowledge-based detection [6,7], (2) behavior specification- based detection [8,9], (3) machine learning-based detection [10,11]. The knowledge-based detection technique identifies the attacks based on the knowledge of specific patterns of misbehavior and system vulnerabilities. However, this approach is ineffective when detecting new unseen cyber-attacks that models have yet to train on before. Behavior specification-based detection tries to detect suspicious behaviors that deviate from normal behaviors. This approach would more effectively detect unknown and unseen cyber-attacks than knowledge-based detection. However, it is challenging to develop models that understand the characteristics of normal behaviors and accurately distinguish them from those caused by cyber-attacks, especially when both normal and attack characteristics are highly similar. Machine learning-based detection is a more generalized approach extended from behavior specification-based detection.

Since it is difficult to obtain labeled attack data in a real-world industrial environment, existing machine learning-based anomaly detection studies have usually focused on an unsupervised learning approach to build a model that detects abnormal behavior using only normal data. A typical anomaly detection system monitors sensors in an ICS and detects sudden changes in the monitored sensor values, which an attack attempt or system fault would cause. Many previous studies demonstrate that deep learning models [12,13,14,15,16] can effectively detect abnormal behaviors of ICS.

Despite the success of anomaly detection models for ICS, it is hard to determine which existing anomaly detection models are the best to apply in an ICS environment because their results were mainly evaluated with their datasets and under different experimental settings and evaluation criteria. For example, Robles-Durazno et al. [17] used F1-score to evaluate model performance, while Pankaj et al. [16] used the true positive rate (TPR) and false positive rate (FPR) to evaluate the performance of models. Therefore, finding practical IDS recommendations for ICS systems takes time and effort. Our work is motivated by the need for comparative evaluation among the state-of-the-art anomaly detection models for ICS. To facilitate direct performance comparisons, we used common datasets and the same experimental settings to compare the performance of state-of-the-art anomaly detection models: InterFusion [13], RANSynCoder [14], GDN [15], LSTM-ED [16], and USAD [12]. We first optimized those models and then compared their performance with two publicly available ICS datasets. Further, we analyzed the effects of various training/testing scenarios on the model performance.

### 1.1. Objectives and Contributions

The objectives of this study are (1) to provide a summary of the technical literature of time series anomaly detection models for ICS, (2) to compare the performance of the state-of-the-art models with two publicly available benchmark datasets, SWaT [18] and HAI [19] datasets, under the same and fair settings to find the best time series anomaly detection model for ICS environments, and (3) analyze the effects of the training dataset sizes to find an appropriate training dataset size minimizing the training overhead. Our main contributions are summarized as follows:We develop a framework to evaluate the performance of anomaly detection models with two public ICS datasets [18,19] and standard evaluation criteria.We provide a comparative evaluation of five promising unsupervised anomaly detection models: InterFusion [13], RANSynCoder [14], GDN [15], LSTM-ED [16], and USAD [12].We implement those five models and optimize them with their hyperparameters.We analyze the effects of the training set size on anomaly detection models and found that most models can achieve a high F1-score with a small portion of the training set, which is comparable with the F1-score when the entire training set is used.

### 1.2. Structure of This Paper

The remainder of this paper is organized as follows: Section 2 provides the background of ICS and the anomaly detection process of the time series dataset. Section 3 discusses previous studies that are related to our work. Section 4 describes the overview of the anomaly detection framework. Section 5 summarizes the five representative anomaly detection models used in our evaluation. Section 6 provides end-to-end performance analysis of the five models with varying training and testing data sizes. We discuss the insights gained from our experimental results in Section 7, followed by the conclusion in Section 8.

## 2. Anomaly Detection in Industrial Control Systems

This section first introduces the concept of ICS and then provides an overview of the time series-based anomaly detection process.

### 2.1. ICS Architecture

ICS manages physical processes in industrial sectors. Although ICS environments are used in large-scale industrial applications (e.g., manufacturers, power plants, and water treatment), the concept of finding optimal ways for the availability of protection and system security is gaining in popularity. Furthermore, recent advances in industrial systems have required integrating Information & Communication Technology (ICT) with the physical process to provide convenient and efficient system management. However, networking and its connectivity to the Internet lead to increased threats within an ICS. Therefore, detecting cyber-attacks in ICS is a challenging task due to the increasing cybersecurity vulnerabilities and diversity of the incidents. To tackle this issue, one basic approach is to continuously monitor measurements of field devices and detect anomaly behaviors in advance to identify the attack source in the real-world ICS.

ICS consists of a control system, a communication network, and field devices, as shown in Figure 1. The field devices are connected with the physical process to collect the raw dataset or execute the control system commands. For instance, sensors measure the water treatment processes to get essential information to manage the system. The programmable logic controller (PLC) reads the sensor datasets as input signals to conduct programmed instructions sent from the control system. Then, the datasets are sent to the control server through the wired or wireless link (e.g., radio tower, power line tower, and satellite tower), which accumulates in a data historian for analysis. Next, the human operator accesses to control server followed by standard protocol to monitor the datasets. Finally, information on the dataset is presented in the human-machine interface (HMI) by querying data historians.

Since the collected dataset contains raw sensor readings of normal and malicious events on the ICS, a continuous analysis to detect anomaly data can help understand the attacker’s behaviors. Therefore, in this paper, we focus on studying the validity of anomaly detection models on datasets collected from real-world ICS devices to improve the security of industrial systems.

### 2.2. Anomaly Detection in ICS

The industrial process executes a series of steps for a given task. Therefore, a collected dataset contains time series characteristics of observations made chronologically. Practically, most time series datasets of ICS have high dimensionality and are collected continuously–called multivariate time series (MTS) data in which logs are collected at every instant from interconnected field devices [20].

While analyzing an MTS dataset of ICS, we have to care for the outliers–abnormal observations collected from sensors and actuators during industrial processes that are predicted as unwanted and deviated from the usual and expected behaviors [21,22]. Generally, there are two main approaches for detecting anomalies in the MTS dataset of ICS. The first approach is based on computing the level of deviation in predicated values at each time instant. Deep neural networks (DNNs) would be proper models for this approach because DNNs can effectively learn non-linear relations of dynamical ICS systems. Using DNNs, we can capture the dynamic nature of ICS datasets and detect outliers or suspicious measurements–often referred to as anomaly detection. The second approach focuses on finding unusual shapes of the MTS dataset; one-class classification models are suitable for this task. However, other methods, such as clustering, would be challenging to handle anomaly detection in MTS dataset due to the high dimensionality of devices [23].

## 3. Related Work

Cybersecurity of ICS attracts a lot of research attention, especially after connecting systems and associated infrastructures to vulnerable networks, which integrate multiple communication protocols and lack proper data detection mechanisms. Here, we highlight studies that showed attackers could infiltrate the network and compromise the whole control system, summarize defenses proposed to defend ICS against cyber-attacks, and demonstrate recent deep anomaly-based models for cyber-attack detection in ICS.

### 3.1. Cyber-Attacks in ICS

Cyber-attacks against ICS have increased in the frequency and sophistication of tactics to avoid detection mechanisms. Firoozjaei et al. [24] demonstrated the adversarial tactics and analyzed the attack mechanisms of six significant real-world ICS cyber incidents in the energy and power industries, namely Stuxnet [25], BlackEnergy [26], Crashoverride [27], Triton, Irongate, and Havex [28]. He provided an evaluation framework for each attack’s threat level of ICS malware and introduced a weighting scheme to rank their influences on ICS. For example, Stuxnet is the world’s first publically known digital weapon that attacked Iran’s nuclear program by targeting ICS and modifying the code running in PLCs to make them deviate from their expected behavior. Triton is the recent malware attack was targeted the control system of an oil and gas plant in Saudi Arabia and damaged the monitoring process of the Schneider Electrics’ Safety Instrumented System (SIS). Besides outside threats, insider threats that originate within targeted systems in ICS are more difficult, destructive, and invisible. Chen et al. [29] evaluated insider threats into three types: non-malicious attacks, malicious attacks, and accidental attacks. All of them can threaten power monitoring systems, abnormal communication systems, and create illegal operations in the industrial control system of power utilities.

### 3.2. Defenses against ICS’s Attacks

There are several surveys that summery defenses from different security aspects and domains in ICS. Giraldo et al. [30] provided a systematic survey of the emerging physics-based defenses in the field of cyber-physical systems (CPS) and proposed a unified taxonomy that helps in developing theoretical foundations, tools, and metrics for different detection methods. Other surveys have focused on defenses of different domains in cyber-physical systems, such as exploring vulnerabilities in smart grids [31,32,33,34], increasing attention to risks in medical devices [35,36,37], and addressing privacy and security issues in general systems [38,39,40,41]. Kayan et al. [42] presented an interesting survey that reviewed the cybersecurity of overall industrial systems and infrastructures, such as Industrial Control Systems (ICS), Industrial cyber-physical systems (ICPSs), Industrial Internet of Things (IIoT), and Industrial Wireless Sensor Networks (IWSNs). However, his survey work focused on proposing a framework to demonstrate current challenges, define systems’ unique characteristics, analyze communication protocols, present an attack taxonomy, and evaluate real-life ICS cyber incidents. To improve cyber-attack detection between smart devices’ communication links in ICS systems, Nedeljkovic et al. [43] proposed a CNN-based method for the design of IDS algorithms over data obtained during normal operations. The method has the ability to dynamically find suitable CNN hyperparameters and thresholds using the training data to define a specific architecture that has good performance in online attack detection.

### 3.3. Deep Anomaly-Based Detection in ICS

Umer et al. [44] presented a survey of methods from machine learning that focused on anomaly detection at the physical level in ICS and IDS at the network level. Wanget et al. [45] addressed the problem of data imbalance in ICS systems which lead to poor performance using traditional ML algorithms. He proposed a transfer learning method by guiding target domain data (poor-to-classify samples) through source domain data (easy-to-classify samples) during training for industrial control anomaly detection. Han et al. [38] showed that the conventional threshold-based anomaly detection approaches of multivariate time series data from networked sensors and actuators in real-world ICS systems suffer from dynamic complexities, especially with the requirement of large amounts of labeled data and continuous monitoring for intrusion incidents. The author proposed a system named MAD-GAN by utilizing Generative Adversarial Networks (GANs) with LSTM to capture the temporal correlation and the entire variable set concurrently, then evaluated MAD-GAN using SWaT and WADI datasets. Wang et al. [46] addressed false data injection cyber-physical attacks (FDIAs) in modern smart grids when generating huge amounts of data. The author proposed an analytical method based on the data-centric paradigm and margin-setting algorithm (MSA) to detect FDIAs. Junejo et al. [47] discussed the issue of late response to an attack, which takes a lot of time to detect the departure of the system from its expected behavior. The author proposed a fast machine learning intrusion detection method based on behaviors of the physical and control components in a modern water treatment system. To enhance the protection of different devices (e.g., sensors, actuators, and controllers) on cyber-physical systems (CPSs), Elgendi et al. [48] proposed a learned (MAPE-K) based model to monitor, analyze, plan, execute, and knowledge against advanced cyber threats and alert users to any abnormalities behavior in an industry environment. To detect attacks on sensors in Cyber-Physical Systems (CPSs), Ahmed et al. [49] proposed a NoisePrint method by creating fingerprints for sensor and process noise during the normal operation of the system. NoisePrint was evaluated using two testbeds, a real-world water treatment (SWaT) and a water distribution (WADI), and achieved an accuracy of higher than 90% against data integrity attacks. It is worth noting that although these studies have been proposed to improve deep anomaly detections to defend against cyber-attacks in ICS, it is hard to claim which model among them is the best to apply and generalize for real-world applications due to their different internal methodologies, diversity in experiments settings, and evaluation approaches. Therefore, our work fills this gap by comprehensively comparing the five common anomaly detection models and giving insights/recommendations to research/industry communities.

## 4. Design of the Time Series Anomaly Detection Framework

This section explains the framework architecture of time series anomaly detection methods in ICS. We assume that an attacker can access field devices through the network and fully understand the target system. Then, the attacker aims to manipulate normal operations to cause a target ICS malfunction and failures.

To fight against such attacks, we can use machine learning models to distinguish normal patterns in input time series measurements from attack behaviors. As shown in Figure 2, the process of unsupervised anomaly detection using deep learning includes three major phases: preprocessing, anomaly scoring, and thresholding. In the preprocessing phase, the raw MTS dataset collected from different field devices (e.g., PLCs, pumps, valves, and motors) may contain noisy readings, missing values, and outliers.

To develop anomaly detection solutions, we assume that the input MTS dataset in ICS is clean measurements, free of outliers, and has no missing values. Therefore, it is necessary to preprocess the raw dataset by applying operations such as normalization and filtering before inputting the data to the models to perform anomaly analysis and detection. However, this stage often results in discarding many time series measurements that fail to meet the criteria of cleaned and scaled data required by models.

Another critical factor in the unsupervised anomaly detection framework is the scoring operation to distinguish abnormal data points from the dataset. The key idea is to define an effective scoring method that maximizes the gap between the normal and attack data (i.e., anomaly), resulting in better prediction performance in terms of false or missed detections. So far, many scoring methods have been proposed in the literature, such as clustering-based, histogram-based, isolation forest, and reconstruction error-based estimations of deep learning autoencoders to detect outliers. For example, for each sample in the test dataset, its score is computed based on how far its characteristics are from the nearest clusters learned from the training dataset. However, one possible way to ensure the effectiveness of a scoring method is to evaluate it on a variety of public MTS datasets and compare it through different performance metrics.

Thresholding is based on the score (numerical value) computed from the previous phase and uses the score as evidence for the final decision (normal or attack). We make a comparison process between the computed score and the given threshold every time to classify a data point. In other words, the threshold value represents the minimum required score value in which the ICS can authorize input measurements. For example, if the input measurements have a score above the threshold, the data will be classified as normal, and the process will be granted. In contrast, suspicious measurements will be classified as anomaly data (attack behaviors) when their score is below the threshold, and hence the process will be denied. In summary, the measurements that provide scores significantly above the threshold are considered strong evidence for a decision of normal operations on ICS and visa-versa. Therefore, the choice of threshold is the crucial goal in this comparison to validate the strength of the decision.

## 5. Anomaly Detection Models

This section introduces the five representative deep neural network models recently developed for anomaly detection and showed a reasonable level of performance. We selected the best-performing anomaly detection models with publicly available source code. The experiment results to evaluate the performance of those models are presented in Section 6.

InterFusion [13]: It is an unsupervised-based anomaly detection and interpretation method that simultaneously investigates two main characteristics of MTS. First, a temporal dependency that describes the periodicity attribute of the patterns within each metric. Second, an inter-metric dependency that models the linear and non-linear relations among all metrics of the MTS at each period. Since previous anomaly detection studies often use deterministic approaches (e.g., prediction-based and reconstruction-based) with only one low-dimensional latent embeddings, they are poor in modeling temporal changes or performing inter-metric anomaly detection. Therefore, the InterFusion detection method tackles this problem by providing a new network architecture design. Specifically, they used Hierarchical Variational Auto-Encoder (HVAE) with jointly two low-dimensional latent variables to explicitly and simultaneously learn both temporal and inter-metric representations to capture normal patterns better. In addition, two-view embedding was designed and added to the network to compress MTS characteristics in both time and metric dimensions. To prove the validity of InterFusion for real-world application in the industrial domains, they evaluated its effectiveness over four different MTS datasets (three existing, which are SWaT [18], WADI [50], and SMD [51], while the fourth, ASD, was newly dataset collected during the InterFusion study). InterFusion is one of the current state-of-the-art anomaly detection methods that provide high and reliable performance for monitoring MTS data for industrial applications.RANSynCoder [14]: It is an unsupervised real-time anomaly detection framework with large multivariate sets. The system architecture is based on multiple encoder-decoders with a pretraining autoencoder. One of the main characteristics of the framework is that the feature bootstrapping aggregation (bagging) [52] is utilized to randomly select the sets of input features and build multiple autoencoders to reconstruct more extensive time series output with smaller sets of input. For anomaly decision, the framework infers anomalies by determining whether each dimension factor of the input is involved between the upper and lower bound of the reconstructed threshold, and when it is not involved, the factor is labeled as an anomaly. To determine the final anomaly decision, the framework applies majority voting to compute the majority label of the dimension factors and finally generate the overall decision. Other than anomaly inference, the framework also provides anomaly localization to identify a significant feature or attribute related to the anomaly alert by measuring the localization score of each feature or attribute.GDN [15]: It is an unsupervised-based anomaly detection method with Graph Attention Network (GAT). GDN makes up for the weak points caused by not explicitly learning with data inter-relationships shown in time series-based features using the graph network method. In addition, each node in the graph structure has sensor embedding, representing each sensor’s unique characteristics. Relationship with the specific sensor is defined to compute the cosine similarity between the target embedding node and other embedding nodes. The similarity score of the embedding vectors utilizes to select the top *k* neighbor nodes with closed relation to a specific node. The graph updates the state of edges and nodes, considering a node’s information with its neighbors in the training step. Each node computes the anomaly score aggregating a single anomaly score for each tick. If the anomaly score deviates from a fixed threshold, the time tick is an anomaly.LSTM-ED [16]: It is an LSTM-based Encoder-Decoder scheme for anomaly detection in multi-sensor MTS datasets. It uses an unsupervised learning method to train its initial network only with unlabeled normal data. Its training involves an encoder learning a fixed-length vector representation of the input MTS dataset, which will help the decoder reconstruct the data accurately. The reconstruction error vector at any future time instance is used to compute the likelihood of anomaly at that point using Maximum Likelihood Estimation (MLE). The intuition behind this is that since the reconstruction error of the trained model is based on normal instances, it produces higher reconstruction errors when anomalous sequences are given. LSTM-ED also provides a threshold mechanism for computed anomaly scores, allowing further tuning of the detection system within a supervised setting to maximize the $F_{\beta }$ score.USAD [12]: It is an unsupervised-based anomaly detection method based on an autoencoder (AE) architecture whose learning is inspired by GANs. USAD provides a new architecture design using two-phase with adversarial training. Training in USAD is done with two autoencoders, $AE_{1}$ and $AE_{2}$. The first autoencoder $AE_{1}$ generates the reconstructed data for the original input dataset using a pair of encoders and decoders. The second autoencoder, $AE_{2}$, intentionally generates anomaly data by adapting adversarial training to the output of the decoder and then giving feedback to the encoder again. In the first phase, the objective is to train each AE to reproduce the input. The second phase aims to train $AE_{2}$ to distinguish the actual data from the data produced from $AE_{1}$. This two-phase training aims to minimize the reconstruction error of *W* and the difference between *W* and the reconstruction output of $AE_{2}$, respectively. In this process, the model learns anomalies that cannot be seen in the original training dataset, obtaining better results. The anomaly score used by USAD for testing is used by multiplying the results obtained after training two autoencoders.

## 6. Evaluation

This section demonstrates the comparative evaluation results of the five anomaly detection models and the effects of training dataset size on detection performance.

### 6.1. Datasets

Deep learning-based anomaly detection approaches need a large dataset to achieve appropriate performance [53]. However, it is difficult to collect the dataset from the ICS that is generally related to critical and fundamental infrastructures. For example, deploying an attack in a real-world ICS can cause severe damage to the ICS and expose its vulnerabilities to attackers. Therefore, collecting a sufficient amount of data from a real-world ICS would be challenging and unacceptable for most ICS. For this reason, the anomaly detection study in ICS uses a dataset collected from the testbed imitating the actual ICS.

In this study, we use two public datasets that are commonly used in anomaly detection research in ICS: Secure Water Treatment (SWaT) [18], and Hardware-In-the-Loop (HIL)-based Augmented ICS (HAI) [19]. Table 1 summarizes the characteristics of the datasets.

SWaT dataset coordinated by Singapore’s Public Utility Board (PUB) represents the overall physical process to control and monitor system behavior. The dataset was collected during 11 days of continuous operations. The first seven days were used to collect sensor data under normal operations, while the remaining four days were used to collect sensor data with attacks. In the SWaT dataset, an attacker intercepts data packets transmitted through EtherNet/IP and Common Industrial Protocol (CIP). The total number of attack types is 41. An attack is labeled 1 on the dataset; a normal operation is labeled 0.

There are several versions of the HAI dataset. Here, we specifically use HAI 21.03, which was released in 2021, where the samples were collected for 19 days from a testbed augmented with a HIL simulator that emulates steam-turbine power generation and pumped-storage hydropower generation—the first five days were used to collect sensor data under normal operations, while the remaining 14 days were used to collect sensor data with attacks. In the HAI dataset, an attacker causes constant errors in the physical measure by manipulating packets. The total number of attack types is 25.

In general, it is important to choose optimal features in designing anomaly detection systems [54]. Therefore, we used XGBoost algorithm [55] to analyze the relative importance of the features for each dataset. Figure 3a shows the importance scores computed for individual features for the SWaT dataset. Flow transmitter is the most important feature with a score of 0.060. Pump controller is the second most important feature with a score of 0.043. Figure 3b shows the importance scores computed for individual features for the HAI dataset. Turbine rotation is the most important feature with a score of 0.096. level switch, water flow, and valve controller would also be important.

Figure 4 and Figure 5 show the four key features of SWaT and HAI datasets, respectively, in distinguishing normal samples from abnormal samples. From those features, we can see clear differences between normal and anomaly distributions. Overall, feature values are scattered across normal samples while located in a smaller range for anomalies.

### 6.2. Evaluation Metrics

It is worth noting that both datasets are imbalanced in which a low percentage of anomalies are collected (11.9% in the SWaT dataset and 2.2% in the HAI dataset). This class imbalance makes it difficult to accurately predict positive classes, misleading the performance score of some metrics. For example, the positive class is 10% of the dataset, and the model can achieve 90% accuracy even if all classes are predicted as negative. For this reason, it is an important issue to determine what performance metrics are used to evaluate the model. We explore 12 recent studies [12,13,14,15,50,51,56,57,58,59,60,61] of IDS research to figure out which performance metrics are commonly used in evaluating anomaly detection models. Most IDS research that applied deep learning models adopt three standard evaluation metrics: Precision, Recall, and $F_{\beta }$-score, as shown in Table 2.

Precision and Recall represent the proportion of false alarms for the number of the true detected anomalies (i.e., True Positives). Prediction focuses on the output of a trained model when tested using new data, regardless of whether the output is correct or wrong. It is sensitive when a model wrongly predicts normal samples as anomalies (i.e., False Positives)–it makes the system deny the normal processes in ICS. On the contrary, Recall focuses on the error among the ground truth anomalies. It is also sensitive when a model mistakenly accepts anomalies from an attacker as normal data (i.e., False Negatives)–this leads to wrongly granting attackers access to the ICS.
(1)$$Precision=\frac{True\;Positive}{True\;Positive+False\;Positive}$$
(2)$$Recall=\frac{True\;Positive}{True\;Positive+False\;Negative}$$

Both metrics are influenced by thresholds used to predict labels. Therefore, measuring the model’s performance may be inappropriate using only Precision or Recall alone. $F_{\beta }$-score is the harmonic mean of Precision and Recall, which is a popular metric for imbalance problem [62,63]. The equation is depicted below:(3)$$F_{\beta }-score=\left(1+\beta^{2}\right)\times \frac{Recall\times Precision}{\beta^{2}\times Recall+Precision}$$

The parameter β is used to adjust the relative importance of Precision and Recall. For example, if the β is higher than 1, $F_{\beta }$-score allocates more weight to Recall. Otherwise, $F_{\beta }$-score gives weight to Precision. The models choose a value of 1 for β, except for the LSTM-NDT using a value of 0.5. It is also important to reduce actual false alarms for availability in an industrial system in real-world ICS. For this reason, we choose the $F_{\beta }$-score to compare the models.

In this study, we consider Precision, Recall, and $F_{\beta }$-score and two false alarm metrics: FalsePositiveRate(FPR) and FalseNegativeRate(FNR) which are popularly used evaluation metrics in intrusion detection systems [39]. FPR is the proportion of false positives when the IDS misidentifies actual normal data as a predicted attack. In contrast, FNR is the proportion of false negatives when the IDS misidentifies actual attack data as a predicted normal. The equations of FPR and FNR are as follows:(4)$$FPR=\frac{False\;Positive}{False\;Positive+True\;Negative}$$
(5)$$FNR=\frac{False\;Negative}{False\;Negative+True\;Positive}$$

### 6.3. Data Preprocessing

We suggest the two data processing techniques as preprocessing steps (data normalization and constant feature exclusion) to process MTS data effectively.

Data normalization (Normalization): It is one of the preprocessing approaches where the values of each feature are either scaled or transformed to make an equal contribution. Because, in an MTS dataset, many attributes have a different scale, data normalization is required.Constant feature exclusion (Feature exclusion): In our work, we exclude constant features, referring to unchanged features, because they would be unnecessary to build a machine learning model. Therefore, a possible strategy is to exclude those features for training. The constant features in each dataset are presented in Table 3.

### 6.4. Model Optimization

Before we compare the performance of the five anomaly detection models for the SWaT and HAI datasets, we first optimize each model with hyperparameters. We focus on the four hyperparameters: epoch, batch size, learning rate, and window size. Table 4 shows the hyperparameter values used to optimize each model.

The epoch is a measure used to update all of the weight vectors for the training set. We tried to find the optimal epoch ranging from 10 to 40 in the model optimization experiment. The batch size means the number of samples for the training set to update the model weight. We tried to find the optimal batch size ranging from 20 to 256. The window size means the length of the input time steps size to forecast the next time based on the feature values. We tried to find the optimal window size ranging from 5 to 1000. The learning rate means the size of a step to reduce the error gradient. We tried to find the optimal learning rate ranging from a small learning rate of 0.0001 to a large learning rate of 0.05. As explained in Section 4, threshold plays a vital role in model performance for anomaly detection. Therefore, we tried to find the best threshold for each model achieving the highest F1-score.

As presented in Section 6.3, we consider two data processing techniques to improve models’ detection accuracy. We found that data normalization is necessary for all models. However, constant feature exclusion is selectively effective depending on the data and model. For the SWaT dataset, the detection accuracy of RANSynCoder and GDN is only improved, while for the HAI dataset, the detection accuracy of GDN is only improved.

### 6.5. Performance Results

To compare the state-of-the-art anomaly detection models’ performance, we implement those models (we used existing code if available) and evaluate them using the same experimental configurations. Table 5 shows the performance results of the five models tested for the SWaT dataset (bold text indicates the highest value in each performance metric). Our implementations achieved F1-score results that are slightly different from the F1-score results presented in the original papers of those models. We use (x% [*y*]) to represent each model’s F1-score of x% presented in the paper [*y*]. We note that our implementations’ (except LSTM-ED) F1-score are slightly lower than the F1-score results presented in the original papers of those models.

For the SWaT dataset, InterFusion produces the best results in F1-score, Recall, and FNR compared to the other four models. Even though the three models (LSTM-ED, GDN, and RANSynCoder) show relatively higher Precision than InterFusion, they achieve significantly lower Recall values, indicating the inferiority of these models to correctly detect anomaly samples and the possibility of false detections for normal samples as anomalies. In contrast, InterFusion is more accurate and shows higher than 90% in both Precision and Recall, indicating that InterFusion is less affected by imbalanced ratios of normal/attack in the dataset than other models. Interestingly, all the anomaly detection models achieve a low FPR, less than or equal to 1.2%, while FNR is over 27%, except for the InterFusion model. We interpret the outperforming of InterFusion over other models because of the model’s well-designed structure with two latent representations, which differs from traditional hierarchical methods. Specifically, ICS produces MTS data with many features (51 columns in SWaT and 79 columns in the HAI) and has non-linear relationships among all of them. Therefore, the InterFusion model architecture was suitable for learning temporal dependencies (intra-metric) and inter-metric relationships to improve MTS’s anomaly detection performance.

Table 6 shows the performance results of the five models tested for the HAI dataset (bold text indicates the highest value in each performance metric). Unlike the SWaT dataset, RANSynCoder produces the best results for the HAI dataset in F1-score, Precision, and FPR. Perhaps, RANSynCoder would be less affected by data imbalance of normal and anomaly samples when considering the HAI dataset (including 2.2% anomalies) is relatively more imbalanced than the SWaT dataset (including 11.9% anomalies). Although InterFusion shows the second-highest performance in F1-score for the HAI dataset, its Recall score is still higher than other models, indicating that it is better to detect attacks. In addition, all the models achieve good Precision scores higher than 74%, while the Recall of GDN and USAD are going worse to less than 50%, indicating that GDN and USAD are ineffective in detecting anomalies when normal and anomaly samples are highly imbalanced.

We also compare the anomaly detection models for the training and testing times for the SWaT and HAI datasets, as shown in Table 7. Training time refers to the time taken to train a model with training samples. Testing time refers to the time taken to perform classification with all testing samples. We implemented models in Python 3 environment. We used a GPU server that consists of a Tesla V100-PCIe with 34GB memory and Intel Xeon(R) E5-2687w v3 @3.10 GHz with 264GB RAM in the experiments.

For the SWaT dataset, the experimental results show that GDN is the fastest model and takes 471 s for training. Because GDN achieves an F1-score higher than 80%, and InterFusion takes over 6000 s for training, we can consider GDN as an alternative to InterFusion when we need to build a model promptly. Although the most efficient model is USAD taking 8 s in testing time, when we consider that the F1-score of USAD is 75.0%, we would not recommend using USAD. Instead, RANSynCoder would be a good candidate model regarding its F1-score (82.7%) and testing time (13 s). In the case of LSTM-ED, model training took significantly longer for the SWaT dataset than the HAI dataset, even if the size of the HAI dataset is considerably larger. This result is due to the larger window size (20 and 4 for SWaT and HAI datasets, respectively) and smaller batch size (40 and 200 for SWaT and HAI, respectively) configured for the SWaT model training, as shown in Table 4.

Similarly, Table 7 shows the training and testing times of the anomaly detection models for the HAI dataset. InterFusion requires a prolonged training time (over 100,000 s) for the HAI dataset.

Moreover, InterFusion takes 8961 s in testing time, significantly higher than the other models’ testing times. We interpret the high time consumption of InterFusion into two aspects. First, it is due to the essential tasks involved in the model’s design of InterFusion, such as the pretraining phase, prefiltering strategy, hierarchical structure, and two-view HVAE embedding for MTS anomaly monitoring. Second, these specifically designed structures require a considerable time when jointly training two stochastic latent variables to capture complex temporal and inter-metric dependencies, especially for large MTS datasets. Hence, we would not recommend using InterFusion with the HAI dataset; a more obvious recommendation would be to use RANSynCoder because it achieves 82.9% F1-score and is significantly faster than InterFusion in training and testing.

### 6.6. Effects of Training Set Size

This section explains how the training data size can affect anomaly detection performance. Deep learning models generally need a large amount of training data to perform well. So naturally, the training time of deep learning models increases proportionally to the training data size. However, an expensive training cost would be unacceptable or unsatisfactory in some situations. Furthermore, it is challenging to collect the ICS data from real-world ICS systems over a long period because the data collection process may expose them to risk.

In this study, we investigate the effects of the training set size on the performance of anomaly detection models by varying the size of the training set. To set up a testing environment with time-series information preserved, we divide the training set into five parts: $p_{1}$, $p_{2}$, $p_{3}$, $p_{4}$, and $p_{5}$, as shown in Figure 6. We train each model with $\left\{p_{1}\right\}$ (20%), $\left\{p_{1},p_{2}\right\}$ (40%), $\left\{p_{1},p_{2},p_{3}\right\}$ (60%), $\left\{p_{1},p_{2},p_{3},p_{4}\right\}$ (80%), and $\left\{p_{1},p_{2},p_{3},p_{4},p_{5}\right\}$ (100%), respectively. For all those configurations, we use the entire testing set in the same manner. The accuracy results are summarized in Table 8 and Table 9, respectively.

Table 8 represents performance variation in InterFusion, RANSynCoder, GDN, LSTM-ED, and USAD models with the training set size. Surprisingly, all the models except USAD show a reasonable F1-score with only 20% of all training samples. We note that those models showed error rates of a very low FPR and relatively high FNR, like the cases when the entire training dataset is used. Unlike the other models, USAD provides poor performance when 20% of all training samples are used.

Similarly, we examined the performance of the five models by varying the size of the training set in the HAI dataset. Table 9 shows the performance results of the models tested. Interestingly, RANSynCoder outperformed InterFusion and showed the highest F1-score when the model was trained using only 40% of all training samples. Regarding FPR and FNR ratios with the HAI dataset, models’ performance improved compared to the SWaT dataset and correctly detected normal and attack samples–this appears from the lower error rates of FNR and FPR.

We also analyze how the training times of the models change with the training set size. Figure 7 shows the training time results with the size of the training set.

The experimental results show that all the models’ training times proportionally increase with the training dataset size. Specifically, InterFusion has the largest variation in the training time compared with other models. In particular, it shows that the training time of the Interfusion for HAI dataset takes longer than three times compared with the SWaT dataset.

## 7. Discussion

This paper provides the performance comparison results of the state-of-the-art anomaly detection methods for ICS. We implemented the five models, InterFusion [13], RANSynCoder [14], GDN [15], LSTM-ED [16], and USAD [12], with two publicly large MTS datasets, SWaT [18] and HAI [19]. Table 10 provides a summary of the five anomaly detection models and their best detection results for both datasets, respectively.

Our experimental results on anomaly detection models suggest the following findings:InterFusion and RANSynCoder show superiority in detection accuracy with the SWaT and HAI datasets. InterFusion might be preferred when Recall is more significant than Precision. However, we note that InterFusion requires expensive costs in training and testing times compared with the other models. Hence, we recommend using RANSynCoder when training and testing times are critical to deploying a model in a target ICS environment.Overall, FNR values are significantly worse than FPR values in all five models. We surmise that each model can sufficiently learn the characteristics of normal data, but it cannot detect new and unseen attacks effectively because the models are well-trained on only normal data. Similarly, our classification result shows relatively higher TN and lower TP when evaluating the performance of each model with both normal and attack samples. We note that identifying attack samples would be more challenging because our model is trained with the normal dataset alone. Therefore, normal samples are relatively well-recognized than attack samples by each model.

Next, we also analyze the effects of dataset size on the performance of the anomaly detection models. Our experimental setup reflects the characteristics of a series of observations arranged chronologically to split the training dataset. Our experimental results on dataset sizes suggest the following findings:We found that a subset (e.g., 40% or 60%) of the entire training set would produce a performance comparable with the entire training dataset. We surmise that this is because SWaT and HAI datasets have some regular periodic normal behaviors. Therefore, a part of the entire training set could be sufficiently used to build an anomaly detection model that achieves reasonable detection accuracy.All the models’ training times proportionally increase with the training set size. Therefore, we can control a model’s training time with the training set’s size when it is important to reduce the training time. Model retraining would be necessary to maintain low error rates over time. To minimize the cost of retraining, we might need to determine the optimal size of the samples for training.

## 8. Conclusions

This work presents a comparative evaluation of the five state-of-the-art time series-based anomaly detection models for ICS. We first focused on finding their optimized models producing the best performance results with two publicly available benchmark datasets, SWaT [18] and HAI [19] datasets, under the same and fair settings.

The evaluation results show that InterFusion and RANSynCoder outperformed the other models’ detection accuracy. For SWaT, InterFusion achieves the highest F1-score of 90.7%, while RANSynCoder achieves the highest F1-score of 82.9% for HAI. Although InterFusion performed well with both datasets, when we consider how expensive the training and testing phases of InterFusion are, we do not recommend using InterFusion. Our more practical recommendation would be to use RANSynCoder regarding the time overhead for training. In addition, we analyzed the effects of the training dataset sizes to investigate whether we could reduce the training overhead and find an appropriate training dataset size. We found that approximately 40% of the entire training set would be sufficient to build a model producing a similar performance compared to the entire training set. These findings help select and design the best model for ICS applications.

However, the detection accuracy of existing anomaly detection models still needs improvement in the real world. A combination of several anomaly detection models could be used together as an ensemble solution rather than just relying on one particular model. Previous studies showed that well-designed ensemble classifiers could effectively produce highly accurate anomaly detection results in other domains. As part of future work, we plan to analyze the performance of various model combinations. Moreover, to alleviate the data imbalance problem, we can use a technique called *data augmentation* to balance the distribution of normal and anomaly samples and avoid performance degradation against new and unseen datasets.

## Figures and Tables

**Figure 1 sensors-23-01310-f001:**
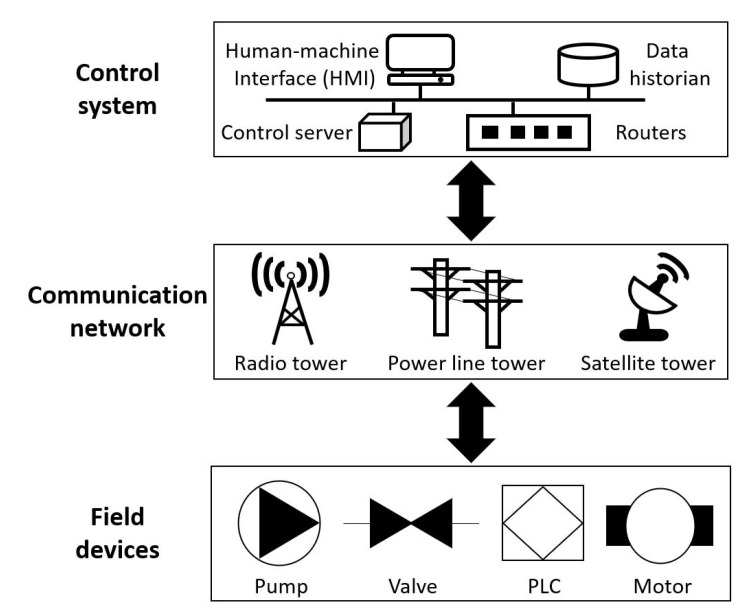
Illustration of ICS structure.

**Figure 2 sensors-23-01310-f002:**
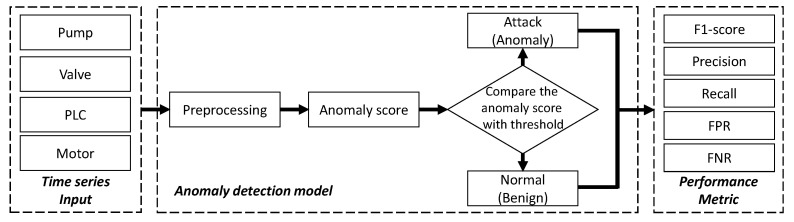
Typical anomaly detection framework using machine learning models in ICS.

**Figure 3 sensors-23-01310-f003:**
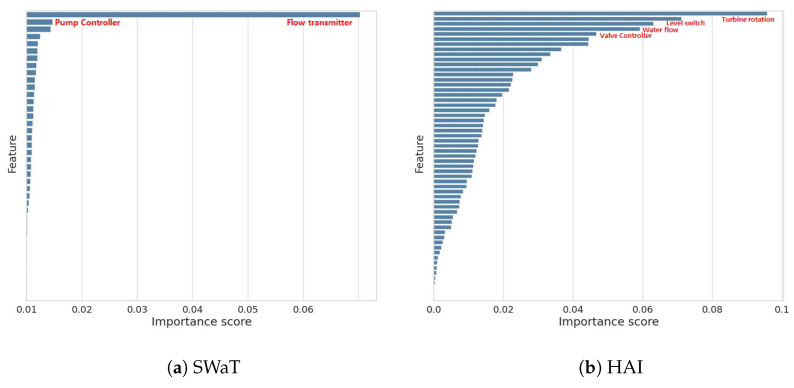
Feature importance score plots based on the SWaT and HAI datasets.

**Figure 4 sensors-23-01310-f004:**
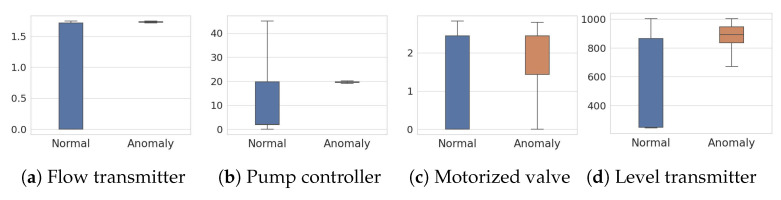
Box plots of normal and anomaly features in the SWaT dataset.

**Figure 5 sensors-23-01310-f005:**
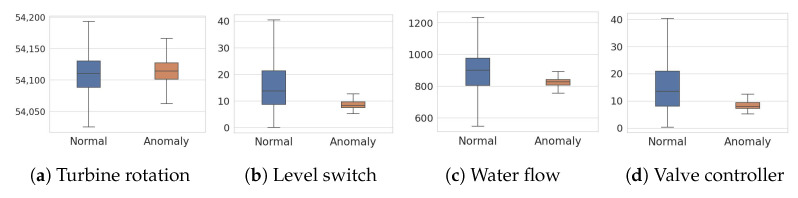
Box plots of normal and anomaly features in the HAI dataset.

**Figure 6 sensors-23-01310-f006:**
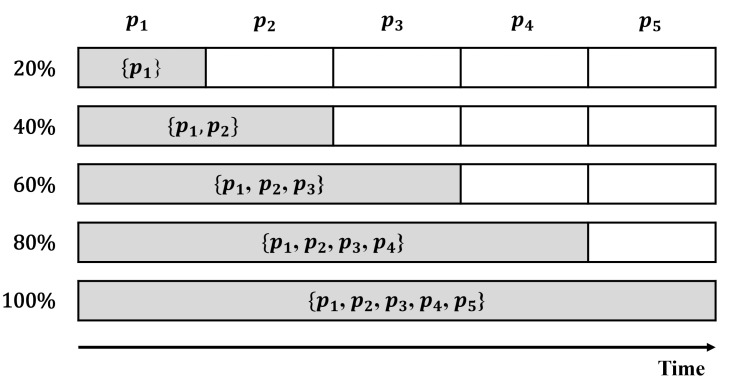
Five training set configurations.

**Figure 7 sensors-23-01310-f007:**
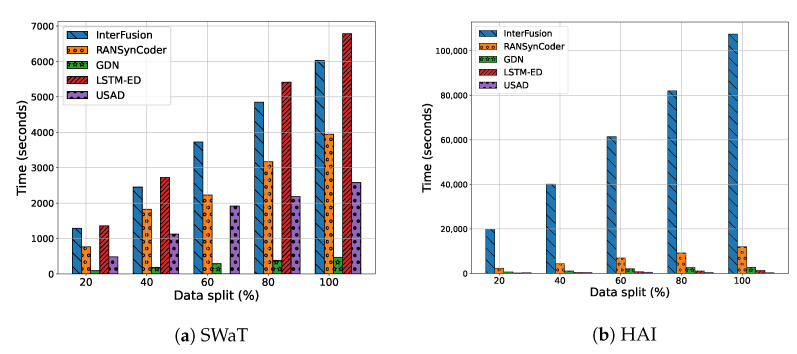
Changes in training times of the anomaly detection models with data split ratio.

**Table 1 sensors-23-01310-t001:** Statistics of evaluation datasets.

Datasets	Features	Training Samples	Testing Samples	Prop. of Anomalies
SWaT	51	322,288	449,919	11.9%
HAI	79	921,603	402,005	2.2%

**Table 2 sensors-23-01310-t002:** Performance metrics of state-of-the-art in time series anomaly detection.

Model	Venue	Precision	Recall	F${}_{\beta }$-score	AUC
InterFusion [13]	KDD	✓	✓	✓	✓
MAD-GAN [50]	ICANN	✓	✓	✓	-
RANSynCoder [14]	KDD	✓	✓	✓	-
GDN [15]	AAAI	✓	✓	✓	-
USAD [12]	KDD	✓	✓	✓	-
DAGMM [56]	ICLR	✓	✓	✓	-
LSTM-NDT [57]	KDD	✓	✓	✓	-
VAEpro [58]	AAAI	-	-	✓	-
DSEBMs [59]	ICML	✓	✓	✓	-
MSCRED [60]	AAAI	✓	✓	✓	✓
OmniAnomaly [51]	KDD	✓	✓	✓	-
IWP-CSO with HNA-NN [61]	KDD	✓	✓	✓	-

**Table 3 sensors-23-01310-t003:** Constant features in the SWaT and HAI datasets.

Datasets	Constant Features
SWaT	P102, P201, P202, P204, P206, P401, P403, P404, P502, P601, P603
HAI	P1_PCV02D, P2_VTR01, P2_VTR04, P4_HT_PS, P1_PP02D
	P4_ST_PS, P1_PP02R, P2_MSD, P2_VTR03, P2_TripEx, P1_STSP
	P2_VTR02, P1_FCV02D, P3_LL, P2_AutoGo, P1_PP01BD, P3_LH
	P1_PP01AD, P1_PP01AR, P2_OnOff, P2_RTR, P2_ManualGo

**Table 4 sensors-23-01310-t004:** Hyperparameter values for model optimization.

Model	SWaT							HAI						
	Epoch	Batch Size	Window Size	Learning Rate	Threshold	Normalization	Feature Exclusion	Epoch	Batch Size	Window Size	Learning Rate	Threshold	Normalization	Feature Exclusion
InterFusion	15	100	30	$1\times 10^{-4}$	−28,972	✓	-	30	100	100	$1\times 10^{-1}$	$-1.29\times 10^{27}$	✓	-
RANSynCoder	10	180	5	$5\times 10^{-2}$	7.366 $\times \;10^{-1}$	✓	✓	10	180	5	$5\times 10^{-2}$	5.798 $\times \;10^{-1}$	✓	-
GDN	10	256	40	$2\times 10^{-2}$	10,776 $\times \;10^{2}$	✓	✓	50	100	60	2 $\times \;10^{-3}$	2095	✓	✓
LSTM-ED	30	40	20	$5\times 10^{-2}$	5892	✓	-	30	200	4	1 $\times \;10^{-3}$	−103.95	✓	-
USAD	10	512	6	$1\times 10^{-2}$	✓	-	10	6	512	1 $\times \;10^{-3}$	8 $\times \;10^{-3}$	✓	-	

**Table 5 sensors-23-01310-t005:** Performance of anomaly detection models for the SWaT dataset.

Model	F1-Score	Precision	Recall	FNR	FPR	TP	FN	FP
InterFusion	**90.7%** (92.8% [13])	91.1%	**90.3%**	**9.7%**	1.2%	**49,309**	**5312**	4799
RANSynCoder	82.7% (84.0% [14])	96.6%	72.3%	27.7%	0.4%	39,511	15,110	1380
GDN	80.6% (81.0% [15])	97.8%	68.5%	31.5%	0.2%	37,403	17,218	836
LSTM-ED	81.2% (76.0% [64])	**98.9%**	68.8%	31.2%	**0.1%**	37,586	17,035	**410**
USAD	75.0% (79.1% [12])	91.6%	63.6%	36.4%	0.8%	34,856	19,940	3208

**Table 6 sensors-23-01310-t006:** Performance of anomaly detection models for the HAI dataset.

Model	F1-Score	Precision	Recall	FNR	FPR	TP	FN	FP
InterFusion	78.9%	74.4%	**83.9%**	**16.1%**	0.6%	**7504**	**1443**	2579
RANSynCoder	**82.9%**	**89.1%**	77.6%	22.4%	**0.2%**	6452	1866	**793**
GDN	59.7%	78.5%	48.3%	54.0%	**0.2%**	4323	4624	1054
LSTM-ED	71.7%	79.1%	65.5%	34.5%	0.4%	5864	3083	1547
USAD	58.8%	76.0%	48.0%	71.3%	**0.2%**	2447	6096	821

**Table 7 sensors-23-01310-t007:** Training and testing times of anomaly detection models in SWaT and HAI datasets.

Model	SWaT		HAI	
	Training Time (s)	Testing Time (s)	Training Time (s)	Testing Time (s)
InterFusion	6032	3563	107,550	8961
RANSynCoder	3945	13	12,100	20
GDN	471	361	2832	482
LSTM-ED	6787	215	1354	79
USAD	2578	8	2653	15

**Table 8 sensors-23-01310-t008:** Performance of anomaly detection models by varying the size of the training set in SWaT.

Model	Training Set Size	F1-Score	Precision	Recall	FNR	FPR	TP	FN	FP
	20%	88.6%	94.1%	83.6%	1.6%	0.7%	45,690	8931	2861
	40%	89.2%	96.9%	82.7%	17.3%	0.4%	45,168	9453	1458
InterFusion	60%	89.8%	94.3%	85.8%	14.2%	0.7%	46,845	7776	5585
	80%	83.2%	88.5%	78.5%	21.5%	1.4%	42,895	11,726	5585
	100%	90.7%	91.1%	90.3%	9.7%	1.2%	49,309	5312	4799
	20%	83.2%	92.1%	75.8%	24.2%	0.9%	41,418	13,203	3546
	40%	81.7%	94.3%	72.2%	27.9%	0.6%	39,411	15,210	2398
RANSynCoder	60%	80.5%	97.8%	68.3%	31.7%	0.2%	37,322	17,299	829
	80%	82.2%	96.3%	71.7%	28.3%	0.4%	39,181	15,440	1489
	100%	82.7%	96.6%	72.3%	27.7%	0.4%	39,511	15,110	1380
	20%	76.9%	96.8%	63.8%	36.2%	0.2%	34,828	19,793	1162
	40%	77.8%	97.8%	64.5%	35.5%	0.1%	35,212	19,409	778
GDN	60%	78.3%	96.7%	65.7%	34.2%	0.3%	35,909	18,712	1206
	80%	78.1%	96.6%	65.6%	34.4%	0.3%	35,843	18,778	1272
	100%	80.6%	97.8%	68.5%	31.5%	0.2%	37,403	17,218	836
	20%	62.8%	82.9%	50.6%	49.4%	1.4%	27,626	26,995	5695
	40%	68.9%	81.3%	59.8%	40.2%	1.8%	32,645	21,978	7492
LSTM-ED	60%	76.2%	98.9%	62.0%	38.0%	0.1%	33,860	20,761	369
	80%	77.4%	99.0%	63.6%	36.4%	0.1%	34,729	19,892	348
	100%	81.2%	98.9%	68.8%	31.2%	0.1%	37,586	17,035	410
	20%	30.6%	19.1%	77.1%	23.0%	45.2%	42,216	12,580	178,658
	40%	74.6%	92.1%	62.7%	37.3%	0.7%	34,343	20,453	2954
USAD	60%	74.3%	92.1%	62.3%	37.7%	0.7%	34,119	20,677	2909
	80%	74.7%	92.5%	62.6%	37.4%	0.7%	34,297	20,499	2792
	100%	75.1%	91.6%	63.7%	36.4%	0.8%	34,856	19,940	3208

**Table 9 sensors-23-01310-t009:** Performance of anomaly detection models by varying the size of the training set in HAI.

Model	Training Set Size	F1-Score	Precision	Recall	FNR	FPR	TP	FN	FP
	20%	75.9%	67.6%	86.5%	13.5%	0.9%	7738	1209	3712
	40%	77.1%	66.4%	91.9%	8.1%	1.1%	8226	721	4158
InterFusion	60%	75.8%	69.4%	83.6%	16.4%	0.8%	7476	1471	3301
	80%	80.2%	74.8%	86.4%	13.5%	0.7%	7734	1213	2610
	100%	78.9%	74.4%	83.9%	16.1%	0.6%	7504	1443	2579
	20%	69.1%	86.7%	57.4%	42.6%	0.2%	4775	3543	731
	40%	88.5%	89.2%	87.8%	12.9%	0.3%	7305	1013	882
RANSynCoder	60%	71.3%	89.7%	59.1%	40.9%	0.2%	4918	3400	563
	80%	70.8%	77.5%	65.1%	34.9%	0.5%	5417	2901	1572
	100%	82.9%	89.1%	77.6%	22.4%	0.2%	6452	1866	793
	20%	31.2%	85.0%	19.1%	80.9%	0.1%	1708	7239	301
	40%	45.5%	63.3%	35.5%	64.5%	0.5%	3178	5769	1846
GDN	60%	53.1%	65.4%	44.4%	55.6%	0.5%	3975	4972	2054
	80%	55.9%	73.3%	45.3%	54.7%	0.4%	4055	4893	1472
	100%	59.7%	78.5%	48.3%	54.0%	0.2%	4323	4624	1054
	20%	15.9%	9.0%	71.3%	28.6%	16.4%	6383	2564	64,573
	40%	72.2%	79.0%	66.4%	33.6%	0.4%	5944	3003	1581
LSTM-ED	60%	71.8%	80.3%	64.9%	35.1%	0.4%	5807	3140	1421
	80%	72.3%	80.0%	65.9%	34.1%	0.4%	5895	3052	1476
	100%	71.7%	79.1%	65.5%	34.5%	0.4%	5864	3083	1547
	20%	60.5%	92.5%	44.9%	73.0%	0.1%	2229	6244	383
	40%	58.6%	94.8%	42.4%	73.8%	0.1%	2231	6312	354
USAD	60%	59.7%	81.5%	47.1%	70.9%	0.1%	2485	6058	608
	80%	61.1%	88.4%	46.7%	71.8%	0.1%	2407	6136	467
	100%	58.8%	76.0%	48.0%	71.3%	0.2%	2447	6096	821

**Table 10 sensors-23-01310-t010:** Summary of five representative anomaly detection models.

Model	Type	F1-Score for SWaT	F1-Score for HAI
InterFusion	VAE	**90.7%**	78.9%
RANSynCoder	AE	82.7%	**82.9%**
GDN	GNN	80.6%	59.7%
LSTM-ED	LSTM	81.2%	71.7%
USAD	AE	75.0%	58.8%

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
