# Peer review of "A Comparative Study of Time Series Anomaly Detection Models for Industrial Control Systems"

_sensors, 2023, doi:10.3390/s23031310_

Round 1
Reviewer 1 Report
1. The abstract of the paper is considered to be the crux of the complete paper, it should address the aim/objective, methods employed and the conclusion of the work done. The abstract presented here lacks in all the above areas and needs to be reworked to provide a better overview of the complete paper.
2. The introduction part of the paper is too short and does not provide a complete understanding and motivational relevance of this study. It needs to be elaborated for the same.
3. Since this is a study article, the literature review part which is of utmost importance in such cases is not there. The author should provide a critical review rather than just an enumeration of various state-of-the-art methods. Also, it would be beneficial to recognize the gaps, pros, and cons of the corresponding techniques to better present the motivation behind this work.
4. The study done by the author in this paper is limited to only a few techniques. There are various other anomaly detection techniques available why those techniques are not taken into account to give a broader view to the study?
5. In the paper, the author has not presented any analytical results to show the accuracy that the proposed system using different techniques.
6. The author has done a study on various anomaly detection methods but has not elaborated on the findings of those studies in his paper which is very pivotal in such types of studies.
7. The result part of the paper is not comprehensive; it should well elaborate on the finding of the study in great detail. Also, the conclusion part needs revision.
8. The author should check for grammatical mistakes in the paper.
Author Response
Thanks for providing constructive review comments for our work. Please see the attached file.

Reviewer 2 Report
The authors have done an excellent job with anomaly detection models for ICS however if the following suggestions are included, the article would be greatly improved.
1. A separate objective section should be included in every research paper so that readers can easily see the article's goal. Authors ought to create a separate section for it.
2. The use of more parametric comparisons by authors is recommended. For instance, confusion matrices are a key component of any IDS system's validity. However, they were seldom mentioned by the authors. As was done in https://doi.org/10.1016/j.cose.2017.04.012, they should compare metrics like "TP, T, FP, and FN rates." (though they have calculated precision & recall using this)
3. Using a neural network algorithm would have been a much better choice, especially given the influence of a bio-inspired algorithm.
In order to express such ideas, they can contrast their work with An Innovative Perceptual Pigeon Galvanized Optimization (PPGO) Based Likelihood Nave Bayes (LNB) Classification Approach for Network Intrusion Detection System.
4. In theory, any distributional disagreement between two datasets might be found using the maximum mean discrepancy (MMD) test. However, it has been demonstrated that the MMD test is blind to adversarial assaults because it was unable to identify the difference between legitimate and malicious data. I query whether natural and adversarial data actually come from different distributions in light of this phenomenon. Authors should express their opinions on this topic.
5. The writers' omission of honeypot anomaly, certainly one of the main issues, is also rather surprising. They really do need to highlight a few statements or pieces of work related to it.
6. Choosing optimal features will play a critical role in any IDS/anomaly research work. Hence authors can refer to the following work https://doi.org/10.1007/s42979-022-01325-4
Author Response

(The authors gave the same response as above.)

Round 2
Reviewer 1 Report
English language and style are fine/minor spell check required
Reviewer 2 Report
The authors have justified my comments and hence the paper can be accepted in its current form.